# Towards a Better Understanding of Variations in Zero-Shot Neural Machine Translation Performance

**Shaomu Tan**     **Christof Monz**
Language Technology Lab
University of Amsterdam
{s.tan, c.monz}@uva.nl

## Abstract

Multilingual Neural Machine Translation (MNMT) facilitates knowledge sharing but often suffers from poor zero-shot (ZS) translation qualities. While prior work has explored the causes of overall low zero-shot translation qualities, our work introduces a fresh perspective: the presence of significant variations in zero-shot performance. This suggests that MNMT does not uniformly exhibit poor zero-shot capability; instead, certain translation directions yield reasonable results. Through systematic experimentation, spanning 1,560 language directions across 40 languages, we identify three key factors contributing to high variations in ZS NMT performance: 1) target-side translation quality, 2) vocabulary overlap, and 3) linguistic properties. Our findings highlight that the target side translation quality is the most influential factor, with vocabulary overlap consistently impacting zero-shot capabilities. Additionally, linguistic properties, such as language family and writing system, play a role, particularly with smaller models. Furthermore, we suggest that the off-target issue is a symptom of inadequate performance, emphasizing that zero-shot translation challenges extend beyond addressing the off-target problem. To support future research, we release the data and models as a benchmark for the study of ZS NMT.[1]

## 1 Introduction

Multilingual Neural Machine Translation (MNMT) has shown great potential in transferring knowledge across languages, but often struggles to achieve satisfactory performance in zero-shot (ZS) translations. Prior efforts have been focused on investigating causes of overall poor zero-shot performance, such as the impact of model capacity (Zhang et al., 2020), initialization (Chen et al., 2022; Gu et al., 2019; Tang et al., 2021; Wang et al., 2021), and how model forgets language labels can affect ZS performance (Wu et al., 2021; Raganato et al., 2021).

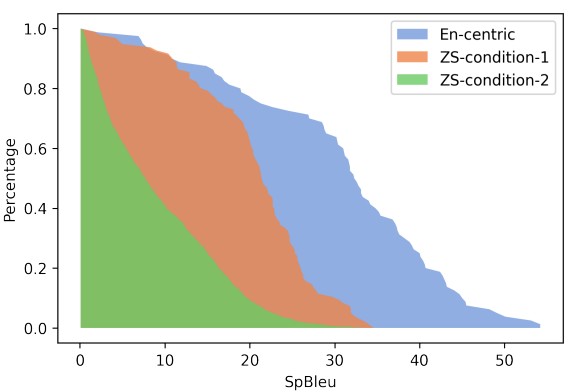

Figure 1: SpBleu Distribution for English-centric and zero-shot directions. The y-axis denotes the percentage of performance surpassing the value on the x-axis. Condition-1 refers to resource-rich ZS directions where the source and target language share linguistic properties, while condition-2 includes other ZS directions.

In contrast, our work introduces a fresh perspective within zero-shot NMT: the presence of high variations in the zero-shot performance. This phenomenon suggests that certain ZS translation directions can closely match supervised counterparts, while others exhibit substantial performance gaps. We recognize this phenomenon holds for both English-centric systems and systems going beyond English-centric data (e.g.: m2m-100 models). This raises the question: which factors contribute to variations in the zero-shot translation quality?

Through systematic and comprehensive experimentation involving 1,560 language directions spanning 40 languages, we identify three key factors contributing to pronounced variations in zero-shot NMT performance: **1)** target side translation capacity, **2)** vocabulary overlap, and **3)** linguistic properties. More importantly, our findings are general regardless of the resource level of languages and hold consistently across various evaluation metrics, spanning word, sub-word, character, and representation levels. Drawing from our findings, we offer potential insights to enhance zero-shot NMT.

---

[1] https://github.com/Smu-Tan/ZS-NMT-Variations/

Our investigation begins by assessing the impact of supervised translation capability on zero-shot performance variations. This is achieved by decomposing the unseen ZS direction (Src→Tgt) into two seen supervised directions using pivot language English. For instance, Src→Tgt can be decomposed into two seen supervised directions: Src→En (source side translation) and En→Tgt (target side translation) for English-centric systems. Our findings show the target side translation quality significantly impacts ZS performance and can explain the variations the most. Surprisingly, the source side translation quality has a very limited impact.

Moreover, our analysis demonstrates the substantial impact of linguistic properties, i.e., language family and writing system, in elucidating the variations in zero-shot performance. Figure 1 highlights this conclusion by showing the much stronger ZS performance of resource-rich ZS directions with similar linguistic properties compared to other directions. Intriguingly, our investigation also shows that the impacts of linguistic properties are more pronounced for smaller models. This suggests that larger models place less reliance on linguistic similarity when engaging in ZS translation, expanding more insights upon prior research about the impact of model capacity on ZS NMT (Zhang et al., 2020).

Furthermore, we found the language pair with higher vocabulary overlap consistently yields better zero-shot capabilities, suggesting a promising future aspect to improve the ZS NMT. In addition, while Zhang et al. (2020) asserts the off-target issue as a primary cause that impairs the zero-shot capability, we conclude that the off-target issue is more likely to be a symptom of poor zero-shot translation qualities rather than the root cause. This is evident by small off-target rates (smaller than 5%) not necessarily resulting in high ZS capabilities.

Lastly, we argue that prior research on zero-shot NMT is limited by focusing only on the 1% of all possible ZS combinations (Aharoni et al., 2019; Pan et al., 2021; Tang et al., 2021) or prioritizing resource-rich language pairs (Yang et al., 2021; Raganato et al., 2021; Chen et al., 2022; Zhang et al., 2020; Qu and Watanabe, 2022). To overcome these limitations, we create the EC40 MNMT dataset for training purposes and utilize multi-parallel test sets for fair and comprehensive evaluations. Our dataset is the first of its kind considering real-world data distribution and diverse linguistic characteristics, serving as a benchmark to study ZS NMT.

## 2 Related Work

### 2.1 MNMT corpus

Current MNMT studies mainly utilize two types of datasets: English-centric (Arivazhagan et al., 2019b; Yang et al., 2021), which is by far the most common approach, and, more rarely, non-English-centric (Fan et al., 2021; Costa-jussà et al., 2022). English-centric datasets rely on bitext where English is either the source or target language, while Non-English-centric ones sample from all available language pairs, resulting in a much larger number of non-English directions. For instance, OPUS100 dataset contains 100 languages with 99 language pairs for training, while the M2M-100 dataset comprises 100 languages covering 1,100 language pairs (2,200 translation directions).

Non-English-centric approaches primarily enhance the translation quality in non-English directions by incorporating additional data. However, constructing such datasets is challenging due to data scarcity in non-English language pairs. In addition, training becomes more computationally expensive as more data is included compared to English-centric approaches. Furthermore, Fan et al. (2021) demonstrates that English-centric approaches can match performance to non-English-centric settings in supervised directions using only 26% of the entire data collection. This suggests that the data boost between non-English pairs has limited impact on the supervised directions, highlighting the promise of improving the zero-shot performance of English-centric systems. Therefore, in this study, we focus on the English-centric setting as it offers a practical solution by avoiding extensive data collection efforts for numerous language pairs.

### 2.2 Understanding Zero-shot NMT

Previous studies have primarily focused on investigating the main causes of overall poor zero-shot performance, such as the impact of model capacity, initialization, and the off-target issue on zero-shot translation. Zhang et al. (2020) found that increasing the modeling capacity improves zero-shot translation and enhances overall robustness. In addition, Wu et al. (2021) shows the same MNMT system with different language tag strategies performs significantly different on zero-shot directions while retaining the same performance on supervised directions. Furthermore, Gu et al. (2019); Tang et al. (2021); Wang et al. (2021) suggest model initializa-

tion impacts zero-shot translation quality. Lastly, Gu et al. (2019) demonstrates MNMT systems are likely to capture spurious correlations and indicates this tendency can result in poor zero-shot performance. This is also reflected in the work indicating MNMT models are prone to forget language labels (Wu et al., 2021; Raganato et al., 2021).

Attention is also paid to examining the relationship between off-target translation and zero-shot performance. Off-target translation refers to the issue where an MNMT model incorrectly translates into a different language (Arivazhagan et al., 2019a). Zhang et al. (2020) identifies the off-target problem as a significant factor contributing to inferior zero-shot performance. Furthermore, several studies (Gu and Feng, 2022; Pan et al., 2021) have observed zero-shot performance improvements when the off-target rate drops.

Our work complements prior studies in two key aspects: **1)** Unlike previous analyses that focus on limited zero-shot directions, we examine a broader range of language pairs to gain a more comprehensive understanding of zero-shot NMT. **2)** We aim to investigate the reasons behind the variations in zero-shot performance among different language pairs and provide insights for improving the zero-shot NMT systems across diverse languages.

## 3  Experiment

### 3.1  EC40 Dataset

Current MNMT datasets pose significant challenges for analyzing and studying zero-shot translation behavior. We identify key shortcomings in existing datasets: **1)** These datasets are limited in the quantity of training sentences. For instance, the OPUS100 (Zhang et al., 2020) dataset covers 100 languages but is capped to a maximum of 1 million parallel sentences for any language pair. **2)** Datasets like PC32 (Lin et al., 2020) fail to accurately reflect the real-world distribution of data, with high-resource languages like French and German disproportionately represented by 40 million and 4 million sentences, respectively. **3)** Linguistic diversity, a critical factor, is often overlooked in datasets such as Europarl (Koehn, 2005) and MultiUN (Chen and Eisele, 2012). **4)** Lastly, systematic zero-shot NMT evaluations are rarely found in existing MNMT datasets, either missing entirely or covering less than 1% of possible zero-shot combinations (Aharoni et al., 2019; Pan et al., 2021; Tang et al., 2021).

To this end, we introduce the EC40 dataset to address these limitations. The EC40 dataset uses and expands OPUS (Tiedemann, 2012) and consists of over 66 million bilingual sentences, encompassing 40 non-English languages from five language families with diverse writing systems. To maintain consistency and make further analysis more comprehensive, we carefully balanced the dataset across resources and languages by strictly maintaining each resource group containing five language families and each family consists of eight representative languages.

EC40 covers a wide spectrum of resource availability, ranging from High(5M) to Medium(1M), Low(100K), and extremely-Low(50K) resources. In total, there are 80 English-centric directions for training and 1,640 directions (including all supervised and ZS directions) for evaluation. To the best of our knowledge, EC40 is the first of its kind for MNMT, serving as a benchmark to study the zero-shot NMT. For more details, see Appendix A.1.

As for evaluation, we specifically chose Ntrex-128 (Federmann et al., 2022) and Flores-200 (Costa-jussà et al., 2022) as our validation and test datasets, respectively, because of their unique multi-parallel characteristics. We combine the Flores200 *dev* and *devtest* sets to create our test set. We do not include any zero-shot pairs in the validation set. These datasets provide multiple parallel translations for the same source text, allowing for more fair evaluation and analysis.

### 3.2  Experimental Setups

**Pre-processing**  To handle data in various languages and writing systems, we carefully apply data pre-processing before the experiments. Following similar steps as prior studies (Fan et al., 2021; Baziotis et al., 2020; Pan et al., 2021), our dataset is first normalized on punctuation and then tokenized by using the Moses tokenizer.[2] In addition, we filtered pairs whose length ratios are over 1.5 and performed de-duplication after all pre-processing steps. All cleaning steps were performed on the OPUS corpus, and EC40 was constructed by sampling from this cleaned dataset.

We then learn 64k joint subword vocabulary using SentencePiece (Kudo and Richardson, 2018). Following Fan et al. (2021); Arivazhagan et al. (2019b), we performed temperature sampling ($T = 5$) for learning SentencePiece subwords to over-

---

[2] https://github.com/moses-smt/mosesdecoder

| | Sacrebleu | | | | Chrf++ | | | | SpBleu | | | | Comet | | | |
|---|---|---|---|---|---|---|---|---|---|---|---|---|---|---|---|---|
| | En→X | X→En | En↔X | ZS | En→X | X→En | En↔X | ZS | En→X | X→En | En↔X | ZS | En→X | X→En | En↔X | ZS |
| *Averaged Performance* | | | | | | | | | | | | | | | | |
| mT-big | 23.1 | 27.5 | 25.3 | 4.9 | 47.1 | 52.6 | 49.9 | 20.5 | 29.9 | 30.6 | 30.2 | 7.3 | 78.4 | 78.3 | 78.3 | 54.7 |
| mBart50 | 22.7 | **29.5** | 26.1 | 6.6 | 46.8 | **53.9** | 50.3 | 23.5 | 29.6 | **32.6** | 31.0 | 9.6 | **80.2** | **80.1** | **80.1** | 58.8 |
| mT-large | **23.6** | 28.7 | **26.1** | **7.0** | **47.6** | 53.3 | **50.4** | **25.3** | **30.5** | 31.8 | **31.1** | **10.1** | 79.2 | 79.0 | 79.1 | **59.5** |
| *Coefficient of Variation (CV)* | | | | | | | | | | | | | | | | |
| mT-big | 0.45 | 0.39 | 0.43 | 0.93 | 0.24 | 0.22 | 0.24 | 0.52 | 0.41 | 0.37 | 0.39 | 0.85 | 0.11 | 0.14 | 0.13 | 0.21 |
| mBart50 | 0.48 | 0.38 | 0.44 | 0.80 | 0.25 | 0.22 | 0.24 | 0.48 | 0.41 | 0.36 | 0.39 | 0.74 | 0.11 | 0.13 | 0.12 | 0.23 |
| mT-large | 0.46 | 0.38 | 0.43 | 0.83 | 0.25 | 0.23 | 0.24 | 0.47 | 0.41 | 0.38 | 0.39 | 0.77 | 0.12 | 0.15 | 0.13 | 0.23 |

Table 1: Average performance scores and coefficient of variation on English-centric and Zero-shot (ZS) directions. The table includes three metrics: Sacrebleu, Chrf++, SpBleu, and Comet. The best performance scores (higher means better) are highlighted in **bold** depending on values before rounding, while the highest CV scores in the coefficient of variation section (higher means more variability) are underlined to highlight high variations.

come possible drawbacks of overrepresenting high-resource languages, which is also aligned with that in the training phase.

**Models** Prior research has suggested that zero-shot performance can be influenced by both model capacity (Zhang et al., 2020) and decoder pre-training (Gu et al., 2019; Lin et al., 2020; Wang et al., 2021). To provide an extensive analysis, we conducted experiments using three different models: Transformer-big, Transformer-large (Vaswani et al., 2017), and fine-tuned mBART50. Additionally, we evaluated m2m-100 models directly in our evaluations without any fine-tuning.

**Training** All training and inference in this work use Fairseq (Ott et al., 2019). For Transformer models, we follow Vaswani et al. (2017) using Adam (Kingma and Ba, 2014) optimizer with $\beta1 = 0.9$, $\beta2 = 0.98$, $\epsilon = 10^{-9}$, warmup steps as 4,000. As suggested by Johnson et al. (2017); Wu et al. (2021), we prepend target language tags to the source side, e.g.: '<2de>' denotes translating into German. Moreover, we follow mBART50 MNMT fine-tuning hyper-parameter settings (Tang et al., 2021) in our experiments. More training and model specifications can be found in Appendix A.2.

**Evaluation** We ensure a comprehensive analysis by employing multiple evaluation metrics, aiming for a holistic assessment of our experiments. Specifically, we utilize four evaluation metrics across various levels: **1)** Chrf++: character level (Popović, 2017), **2)** SentencePieceBleu (SpBleu): tokenized sub-word level (Goyal et al., 2022), **3)** Sacrebleu: detokenized word level (Post, 2018)), and **4)** COMET: representation level (Rei et al., 2020). For the Comet score, we use the wmt22-comet-da model(Rei et al., 2022).

While we acknowledge that metrics like Sacrebleu may have limitations when comparing translation quality across language pairs, we believe that consistent findings across all these metrics provide more reliable and robust evaluation results across languages with diverse linguistic properties. For Comet scores, we evaluate the supported 35/41 languages. As for beam search, we use the beam size of 5 and a length penalty of 1.0.

## 4 Variations in Zero-Shot NMT

Table 1 presents the overall performance of three models for both English-centric and zero-shot directions on four metrics. It is evident that all models exhibit a substantial performance gap between the supervised and zero-shot directions. Specifically, for our best model, the zero-shot performances of Sacrebleu and SpBleu are less than one-third compared to their performance in supervised directions, which highlights the challenging nature of zero-shot translation. In addition, compare the results of mT-big and mT-large, we observe that increasing the model size can benefit zero-shot translation, which aligns with previous research (Zhang et al., 2020). Furthermore, we show that while the mBART50 fine-tuning approach shows superior performance in Src→En directions, it consistently lags behind in En→Tgt and zero-shot directions.

**Does pre-training matter?** Gu et al. (2019); Tang et al. (2021); Wang et al. (2021) have shown that pre-trained seq2seq language models can help alleviate the issue of forgetting language IDs often observed in Transformer models trained from scratch, leading to improvements in zero-shot performance. However, our results show an interesting finding: When the MNMT model size matches that of the pre-trained model, the benefits of pre-

| | Sacrebleu | | | Chrf++ | | |
|---|---|---|---|---|---|---|
| | En-cent. | Sup. | ZS | En-cent. | Sup. | ZS |
| Averaged Performance | | | | | | |
| mT-large | 27.7 | 11.9 | 6.2 | 52.5 | 34.1 | 23.2 |
| m2m100-419m | 24.5 | 15.7 | 7.4 | 49.2 | 39.3 | 25.8 |
| m2m100-1.2b | **29.0** | **18.9** | **9.9** | **52.9** | **42.8** | **29.5** |
| Coefficient of Variation (CV) | | | | | | |
| mT-large-418m | 0.36 | 0.86 | 0.96 | 0.20 | 0.38 | 0.50 |
| m2m100-419m | 0.45 | 0.60 | 1.05 | 0.27 | 0.36 | 0.57 |
| m2m100-1.2b | 0.43 | 0.57 | 0.96 | 0.25 | 0.33 | 0.52 |

Table 2: Evaluation of m2m100 models on our benchmark. We consider English-centric, supervised and zero-shot directions in this table according to m2m100. This means many "supervised" directions are actually unseen for our mT-large system.

training on zero-shot NMT become less prominent. This result is consistent for both seen and unseen languages regarding mBart50, see Appendix A.4. Our observation aligns with previous claims that the mBART model weights can be easily washed out when fine-tuning with large-scale data on supervised directions (Liu et al., 2020; Lee et al., 2022).

**Quantifying variation**  We identify the higher variations that exist in zero-shot translation performance than supervised directions by measuring the Coefficient of Variation ($CV = \frac{\sigma}{\mu}$) (Everitt and Skrondal, 2010). The CV metric is defined as the ratio of the standard deviation $\sigma$ to the mean $\mu$ of performance, which is more useful than purely using standard deviation when comparing groups with vastly different mean values.

As shown in Table 1, we find substantially higher CV scores in the zero-shot directions compared to the supervised ones, with an average increase of around 100% across all models and metrics. This observation highlights that zero-shot performance is much more prone to variations compared to the performance of supervised directions. This raises the question: What factors contribute to the significant variations observed in zero-shot performance?

**Exploring the Role of Non-English-Centric Systems**  Training with non-English language pairs has shown promise in improving zero-shot performance (Fan et al., 2021). To delve deeper into this aspect, we evaluate m2m100 models directly without further finetuning on our benchmark test set because our goal is to investigate whether the high variations in the zero-shot performance phenomenon hold for non-English-centric models.

Our analysis consists of English-centric (54), supervised (546), and zero-shot (860) directions,

which are determined by the training settings of m2m100. The results in Table 2 yield two important observations. Firstly, significant performance gaps exist between supervised and zero-shot directions, suggesting that the challenges of zero-shot translation persist even in non-English-centric systems. More importantly, our finding of considerable variations in zero-shot NMT also holds for non-English-centric systems.

## 5 Factors in the Zero-Shot Variations

We investigate factors that might contribute to variations in zero-shot directions: **1)** English translation capability **2)** Vocabulary overlap **3)** Linguistic properties **4)** Off-target issues. For consistency, we use our best model (mT-large) in the following analyses, unless mentioned otherwise. For simplicity, we denote the zero-shot direction as Src→Tgt throughout the following discussion, where Src and Tgt represent the Source and Target language respectively. In this chapter, we present all analysis results using SpBleu and provide results based on other metrics in the Appendix for reference.

| | | | Target | | | |
|---|---|---|---|---|---|---|
| | | En | High | Med | Low | e-Low | Avg |
| Source | En | - | 37.53 | 38.90 | 27.11 | 18.47 | 30.50 |
| | High | 36.32 | 16.50 | 17.25 | 8.45 | 3.77 | 11.38 |
| | Med | 36.26 | 16.48 | 17.01 | 8.79 | 4.18 | 11.61 |
| | Low | 30.97 | 13.12 | 13.61 | 6.70 | 3.25 | 9.17 |
| | e-Low | 23.65 | 11.07 | 12.02 | 6.96 | 3.59 | 8.41 |
| | Avg | 31.80 | 14.18 | 14.97 | 7.72 | 3.70 | 10.12 |

Table 3: Resource-level analysis based on SpBleu (we provide analyses based on other metrics in appendix A.5.1). We include both English-centric (shaded blue) and zero-shot (shaded red) directions. Avg→Avg denotes averaged zero-shot SpBleu score.

### 5.1 English translation capability

We first hypothesize that data size, which is known to play a crucial role in supervised training, may also impact zero-shot capabilities. We categorize data resource levels into four classes and examine their performance among each other as shown in Table 3. English-centric results are also included for comparison. Our findings indicate that the resource level of the target language has a stronger effect on zero-shot translations compared to that of the source side. This is evident from the larger drop in zero-shot performance (from 14.18 to 3.70) observed when the target data size decreases, as opposed to the source side (from 11.38 to 8.41).

**Setup** To further quantify this observation, we conducted correlation and regression analyses, see Table 4, following Lauscher et al. (2020) to analyze the effect of data size and English-centric performance. Specifically, we calculate both Pearson and Spearman for correlation, and use Mean absolute error (MAE) and root mean square error (RMSE) for regression. We use data size after temperature sampling in Src→En and En→Tgt directions, as well as the corresponding performances as features.

**Results** Three key observations can be made from these results: **1)** The factors on the target side consistently exhibit stronger correlations with the zero-shot performance, reinforcing our conclusions from the resource-level analysis in Table 3. **2)** The English-centric performance feature demonstrates a greater R-square compared to the data size. This conclusion can guide future work to augment out-of-English translation qualities, we further expand it in the section 6. **3)** We also observe that the correlation alone does not provide a comprehensive explanation for the underlying variations observed in zero-shot performance by visualizing the correlation (Figure 2).

| Metrics | Features | Data-size[†] | | En-centric perf. | |
|---|---|---|---|---|---|
| | | Src_size | Tgt_size | Src→En | En→Tgt |
| Correlation | Pearson | 0.16 | 0.53 | 0.41 | 0.68 |
| | Spearman | 0.18 | 0.61 | 0.41 | 0.69 |
| Regression | R-square | 32.54% | | 61.34% | |
| | MAE | 5.47 | | 4.70 | |
| | RMSE | 6.62 | | 5.59 | |

Table 4: Analysis of zero-shot performance considering data size and English-centric performance based on SpBleu. Data-size[†] is after the temperature sampling as it represents the actual size of the training set.

## 5.2 The importance of Vocabulary Overlap

Vocabulary overlap between languages is often considered to measure potential linguistic connections such as word order (Tran and Bisazza, 2019), making it a more basic measure of similarity in surface forms compared to other linguistic measurements such as language family and typology distance (Philippy et al., 2023). Stap et al. (2023) also identifies vocabulary overlap as one of the most important predictors for cross-lingual transfer in MNMT. In our study, we investigate the impact of vocabulary sharing on zero-shot NMT.

We build upon the measurement of vocabulary overlap proposed by Wang and Neubig (2019)

and modify it as follows: $Overlap = \frac{|V_{Src} \cap V_{Tgt}|}{|V_{Tgt}|}$, where $V_{Src}$ and $V_{Tgt}$ represent the vocabularies of the source (Src) and target (Tgt) languages, respectively. This measurement quantifies the proportion of subwords in the target language that is shared with the source language in the zero-shot translation direction.

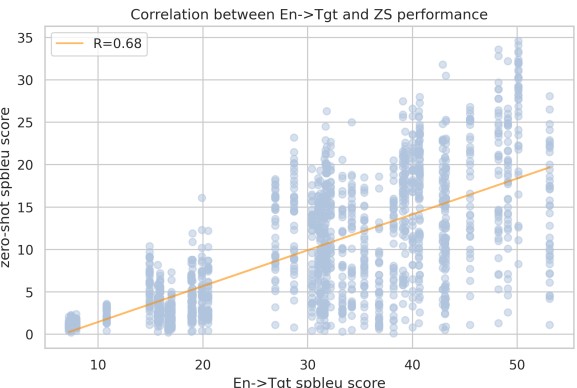

Figure 2: Correlation between En→Tgt SpBleu and zero-shot (All↔Tgt) SpBleu. Each faded blue point denotes the performance of a single zero-shot direction based on SpBleu. R=0.68 indicates the Pearson correlation coefficient (see Table 4 for more details).

**Setup** We first investigate the correlation between the vocabulary overlap and zero-shot performance. As noted by Philippy et al. (2023), vocabulary overlap alone is often considered insufficient to fully explain transfer in multilingual systems. We share this view, particularly in the context of multilingual translation, where relying solely on vocabulary overlap to predict zero-shot translation quality presents challenges. Hence, we incorporate the extent of the vocabulary overlap factor into our regression analysis with English translation performance in section 5.1.

**Results** As shown in Table 5. The results indicate that considering the degree of overlap between the source and target languages further contributes to explaining the variations in zero-shot performance. Importantly, this pattern holds true across different model capacities, and it shows more consistent results than linguistic features such as script and family. We recognize this conclusion can promote to future investigation on how to improve the zero-shot NMT. For example, encouraging greater cross-lingual transfer via better vocabulary sharing by leveraging multilingual dictionaries, or implicitly learning multilingual word alignments via multi-source translation, we leave them to future work.

| ID | Features | R-square | MAE | RMSE |
|---|---|---|---|---|
| | mT-big | | | |
| 1 | En_performance | 45.63% | 4.03 | 4.88 |
| 2 | 1 + Vocab-Sim | 63.42% | 3.70 | 4.77 |
| 3 | 2 + Linguistic-features | 81.17% | 3.42 | 4.37 |
| | mT-large | | | |
| 4 | En_performance | 61.34% | 4.70 | 5.59 |
| 5 | 4 + Vocab-Sim | 79.75% | 3.76 | 4.84 |
| 6 | 5 + Linguistic-features | 81.75% | 3.67 | 4.75 |

Table 5: Prediction of Zero-Shot Performance using En-Centric performance, vocabulary overlap, and linguistic properties. We present the result based on SpBleu in this table.

## 5.3 The impact of Linguistic Properties

Previous work on cross-lingual transfer of NLU tasks, such as NER and POS tagging, highlights that transfer is more successful for languages with high lexical overlap and typological similarity (Pires et al., 2019) and when languages are more syntactically or phonologically similar (Lauscher et al., 2020). In the context of multilingual machine translation, although it is limited to only validating four ZS directions, Aharoni et al. (2019) has empirically demonstrated that the zero-shot capability between close language pairs can benefit more than distant ones when incorporating more languages.

Accordingly, we further extend this line of investigation by examining linguistic factors that may impact zero-shot performance in MNMT. We measure the role of two representative linguistic properties, namely language family and writing system, in determining the zero-shot performance. The specific information on linguistic properties of each language can be found in Appendix A.1.

**Setup** To examine the impact of linguistic properties on zero-shot performance, we specifically evaluate the performance in cases where: 1) source and target language belong to the same or different family and 2) source and target language use the same or different writing system. This direct comparison allows us to assess how linguistic similarities between languages influence the effectiveness of zero-shot translation.

**Results** To provide a fine-grained analysis, we examine this phenomenon across different resource levels for the target languages, as shown in Table 6. The results reveal a significant increase in zero-shot performance when the source and target languages share the same writing system, irrespective of the

resource levels. Additionally, we observe that the language family feature exhibits relatively weaker significance as shown in Appendix A.5.3. To further quantify the effect of these linguistic properties on ZS NMT, we conduct a regression analysis, see Table 5. Our findings highlight their critical roles in explaining the variations of zero-shot performance.

| | Tgt resource | | | |
|---|---|---|---|---|
| | eLow | Low | Med | High |
| If Src and Tgt in the same Language Family | | | | |
| No | 2.12 | 4.82 | 9.77 | 9.43 |
| Yes | **3.14**[*] | **7.69**[*] | **13.16**[*] | **12.88**[*] |
| If Src and Tgt use the same Writing System | | | | |
| No | 1.58 | 3.97 | 9.31 | 8.67 |
| Yes | **3.21**[*] | **8.13**[*] | **11.71**[*] | **12.68**[*] |

[*] represents $p <= 0.05$

Table 6: The impact of linguistic properties on zero-shot performance (we use mT-large and SpBleu here for an example). We conduct Welch's t-test to validate if one group is significantly better than another. The detailed table, including the impact of X resource, can be found in Appendix A.5.3.

Furthermore, our analysis reveals interesting findings regarding the effect of linguistic properties considering the model size. As shown in Table 5, we observed that the contribution of linguistic features is more pronounced for the smaller model, i.e., mT-big. While the larger model tends to place more emphasis on English-centric performance. This suggests that smaller models are more susceptible to the influence of linguistic features, potentially due to their limited capacity and generalization ability. In contrast, larger models exhibit better generalization capabilities, allowing them to rely less on specific linguistic properties.

## 5.4 The role of Off-Target Translations

Previous work (Gu and Feng, 2022; Pan et al., 2021) have demonstrated a consistent trend, where stronger MNMT systems generally exhibit lower off-target rates and simultaneously achieve better zero-shot BLEU scores. To further investigate this, we analyze the relationship between off-target rates and different levels of zero-shot performance.

**Setup** We adopt the off-target rate measurement from Yang et al. (2021) and Costa-jussà et al. (2022) using fasttext (Joulin et al., 2016) to detect if a sentence is translated into the correct language.

**Results** While Zhang et al. (2020) identifies the off-target issue as a crucial factor that contributes

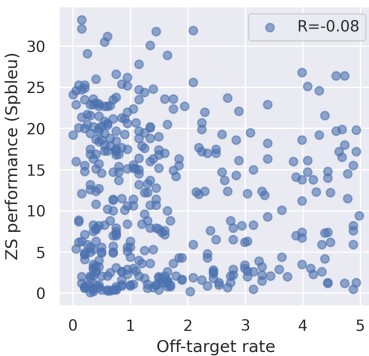

Figure 3: Correlation between off-target rate and zero-shot performance (SpBleu). R represents the Spearman correlation coefficient. We focus on directions where the off-target rate is considerably low (less than 5%). Results based on other metrics can be found in A.5.5.

to poor zero-shot results. However, our analysis, as illustrated in Figure 3, suggests that the reasons for poor zero-shot performance go beyond just the off-target issue. Even when the off-target rate is very low, e.g., less than 5% of sentences being off-target, we still observe a wide variation in zero-shot performance, ranging from very poor (0.1 SpBleu) to relatively good (34.6 SpBleu) scores. Based on these findings, we conclude that the off-target issue is more likely to be a symptom of poor zero-shot translation rather than the root cause. This emphasizes that translating into the correct language cannot guarantee decent performance.

## 6 From Causes to Potential Remedies

In this section, we summarize our findings and offer insights, building upon the previous observations.

**Enhance target side translation**  We identified that the quality of target side translation (En→Tgt) strongly influences the overall zero-shot performance in an English-centric system. To this end, future research should explore more reliable approaches to enhance the target side translation capability. One practical promising direction is the use of back-translation (Sennrich et al., 2016) focusing more on improving out-of-English translations. Similarly, approaches like multilingual regularization, sampling, and denoising are worth exploring to boost the zero-shot translation directions.

**Focus more on distant pairs**  We recognize that distant language pairs constitute a significant percentage of all zero-shot directions, with 61% involving different scripts and 81% involving different language families in our evaluations. Our analysis reveals that, especially with smaller models, distant pairs exhibit notably lower zero-shot performance compared to closer ones. Consequently, enhancing zero-shot performance for distant pairs is a key strategy to improve overall capability. An unexplored avenue for consideration involves multi-source training (Sun et al., 2022) using Romanization (Amrhein and Sennrich, 2020), with a gradual reduction in the impact of Romanized language.

**Encourage cross-lingual transfer via vocabulary sharing**  Furthermore, we have consistently observed that vocabulary overlap plays a significant role in explaining zero-shot variation. Encouraging greater cross-lingual transfer and knowledge sharing via better vocabulary sharing has the potential to enhance zero-shot translations. Previous studies (Wu and Monz, 2023; Maurya et al., 2023) have shown promising results in improving multilingual translations by augmenting multilingual vocabulary sharing. Additionally, cross-lingual pre-training methods utilizing multi-parallel dictionaries have demonstrated improvements in word alignment and translation quality (Ji et al., 2020; Pan et al., 2021).

## 7 Conclusion

In this work, we introduce a fresh perspective within zero-shot NMT: the presence of high variations in the zero-shot performance. We recognize that our investigation of high variations in zero-shot performance adds an important layer of insight to the discourse surrounding zero-shot NMT, which provides an additional perspective than understanding the root causes of overall poor performance in zero-shot scenarios.

We first show the target side translation quality significantly impacts zero-shot performance the most while the source side has a limited impact. Furthermore, we conclude higher vocabulary overlap consistently yields better zero-shot performance, indicating a promising future aspect to improve zero-shot NMT. Moreover, linguistic features can significantly affect ZS variations in the performance, especially for smaller models. Additionally, we emphasize that zero-shot translation challenges extend beyond addressing the off-target problem.

We release the EC-40 MNMT dataset and model checkpoints for future studies, which serve as a benchmark to study zero-shot NMT. In the future, we aim to investigate zero-shot NMT from other views, such as analyzing the discrepancy on the representation level.

## Limitations

One limitation of this study is the over-representation of Indo-European languages in our dataset, including languages in Germanic, Romance, and Slavic sub-families. This could result in non-Indo-European languages being less representative in our analysis. Additionally, due to data scarcity, we were only able to include 5 million parallel sentences for high-resource languages. As a result, the difference in data size between high and medium-resource languages is relatively small compared to the difference between medium and low-resource languages (which is ten times). To address these limitations, we plan to expand the EC40 dataset in the future, incorporating more non-Indo-European languages and increasing the data size for high-resource languages.

## Broader Impact

We collected a new multilingual dataset (EC40) from OPUS, which holds potential implications for the field of multilingual machine translation. The EC40 dataset encompasses a diverse range of languages and language pairs, offering researchers and developers an expanded pool of data for training and evaluating translation models. It also serves as a benchmark for enabling fair comparisons and fostering advancements in multilingual translation research. Recognizing the inherent risks of mistranslation in machine translation data, we have made efforts to prioritize the incorporation of high-quality data, such as the MultiUN (Chen and Eisele, 2012) dataset (translated documents from the United Nations), to enhance the accuracy and reliability of the EC40 dataset. By sharing the EC40 dataset, we aim to contribute to the promotion of transparency and responsible use of machine translation data, facilitating collaboration and driving further progress in multilingual machine translation research.

## Acknowledgments

This research was funded in part by the Netherlands Organization for Scientific Research (NWO) under project number VI.C.192.080. We would like to thank Di Wu, Yan Meng, David Stap, and Baohao Liao for their useful comments and discussions. We would also like to thank the reviewers for their feedback.

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

## A  Appendices

### A.1  Dataset Statistics

We list the details of the EC40 dataset in Table 7. Overall, EC40 is an English-centric multilingual machine translation dataset containing over 66 million sentences including 41 languages (together with English). EC40 is more profound in the total number of languages and in the balance of language family and writing systems. Specifically, for each language family, we include 8 representative languages across different resources.

Moreover, we set the number of sentences the same for each resource level, e.g.: all High-resource Languages have 5M sentences. Note: we list precise numbers in the table instead of approximate ones, for instance, 5M denotes exactly 5,000,000 number of sentences after pre-processing. We use ISO 639-1[3] in this table. We follow Flores-200 to label the writing system classes and use WALS (Dryer and Haspelmath, 2013) to label the language family for languages in our dataset.

### A.2  Training and Model specification

| Model | Num | Encoder layers | Decoder layers | Emb | FFN | Head | Vocab size |
|-------|-----|----------------|----------------|-----|-----|------|------------|
| mT-big | 241M | 6 | 6 | 1024 | 4096 | 16 | 64k |
| mT-large | 418M | 12 | 12 | 1024 | 4096 | 16 | 64k |
| mBart50 FT | 610M | 12 | 12 | 1024 | 4096 | 16 | 250k |

Table 8: Model specification

We also show the model specifications of mTransformer-big, mTransformer-large, and mBart50 Fine-tuning in the Table 8. It is worth noting that mBart50 utilizes vocabulary larger than our trained-from-scratch models. Furthermore, we adopt Vaswani et al. (2017) to set up the learning rate as 5e-4 with 4000 warmup steps and label smoothing of 0.1.

To keep the consistency of learning Sentence-Piece vocabulary, we also used temperature sampling ($T = 5$) for training all models. We trained all models (including mBart50 FT) with 4 NVIDIA A6000 GPUs for a maximum of 200k updates. For larger models, we set the total max tokens as 215,040 using gradient accumulation to stimulate the large batch-size training in Tang et al. (2021).

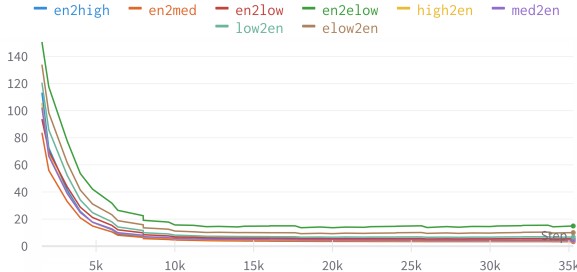

Figure 4: perplexity curves on English-centric directions on our test-set.

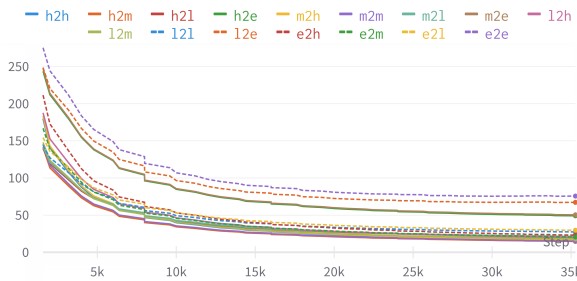

Figure 5: perplexity curves on zero-shot directions on our test-set. h, m, l, e denote High, Medium, Low, and extremely-Low resource levels, respectively.

### A.3  Validation of Spurious Correlation

To ensure that our model does not inadvertently capture spurious correlations during training, we conduct a validation process by visualizing the perplexity curves for both English-centric and zero-shot directions as proposed by (Gu et al., 2019). It is important to note that these curves are solely used for visualization purposes and are not used as criteria for early stopping. Our early stopping criterion for training is based solely on the validation perplexity, and we only consider English-centric directions in this regard.

In Figure 4 and Figure 5, we present the perplexity curves for English-centric and zero-shot directions, respectively. We observe that the perplexities for the zero-shot directions gradually decrease during training, indicating that the model is learning and improving its translation performance on those directions. Importantly, no significant overfitting patterns are observed in the zero-shot perplexity curves. Instead, the decreasing perplexities on zero-shot directions suggest that the model is effectively learning the underlying patterns and generalizing its translation capabilities to unseen language pairs.

---

[3] https://en.wikipedia.org/wiki/ISO_639-1

| | Germanic | | | Romance | | | Slavic | | | Indo-Aryan | | | Afro-Asiatic | | |
|---|---|---|---|---|---|---|---|---|---|---|---|---|---|---|---|---|
| | ISO | Language | Script | ISO | Language | Script | ISO | Language | Script | ISO | Language | Script | ISO | Language | Script |
| High | de | German | Latin | fr | French | Latin | ru | Russian | Cyrillic | hi | Hindi | Devanagari | ar | Arabic | Arabic |
| (5m) | nl | Dutch | Latin | es | Spanish | Latin | cs | Czech | Latin | bn | Bengali | Bengali | he | Hebrew | Hebrew |
| Med | sv | Swedish | Latin | it | Italian | Latin | pl | Polish | Latin | kn | Kannada | Devanagari | mt | Maltese | Latin |
| (1m) | da | Danish | Latin | pt | Portuguese | Latin | bg | Bulgarian | Cyrillic | mr | Marathi | Devanagari | ha | Hausa* | Latin |
| Low | af | Afrikaans | Latin | ro | Romanian | Latin | uk | Ukrainian | Cyrillic | sd | Sindhi | Arabic | ti | Tigrinya | Ethiopic |
| (100k) | lb | Luxembourgish | Latin | oc | Occitan | Latin | sr | Serbian | Latin | gu | Gujarati | Devanagari | am | Amharic | Ethiopic |
| eLow | no | Norwegian | Latin | ast | Asturian | Latin | be | Belarusian | Cyrillic | ne | Nepali | Devanagari | kab | Kabyle* | Latin |
| (50k) | ic | Icelandic | Latin | ca | Catalan | Latin | bs | Bosnian | Latin | ur | Urdu | Arabic | so | Somali | Latin |

Table 7: Details of Our EC40 Multilingual Machine Translation Dataset. Numbers in the table represent the number of sentences, for example, 5m denotes exactly 5,000,000 number of sentences. Two exceptions are Hausa and Kabyle, where their data-size are 334k (334,000) and 18k (18,448) respectively.

## A.4 mBART50 Performance comparison

We show performance comparison results on English-centric and ZS directions in Figure 6, 7, and 8 categorized by both seen and unseen languages. It is clear that fine-tuned mBart50 outperforms the mT-large model on most of X→En directions, but lags behind in En→X and zero-shot directions.

Furthermore, our conclusions are more comprehensive and reliable due to two factors: 1) compare to Tang et al. (2021), our evaluation set encompasses multi-parallel sentences, allowing for performance assessment across various language pairs, including low-resource directions. 2) compare to Wang et al. (2021), we employ the Transformer-large model, enabling a fairer comparison to mBART50 in terms of model size.

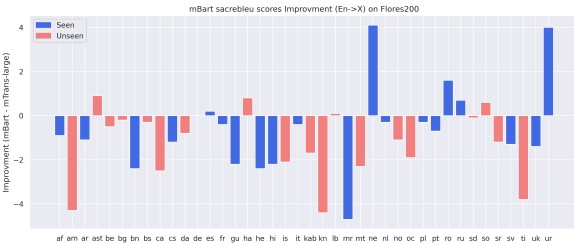

Figure 6: Performance comparison between fine-tuned mBart50 and mT-large on En2X directions.

## A.5 Additional Experiment Results based on other metrics

We ensure a comprehensive analysis by employing multiple evaluation metrics, aiming for a holistic assessment of our experiments. In the paper, we have already shown the results based on the SpBleu, thus, we provide results for all analyses based on

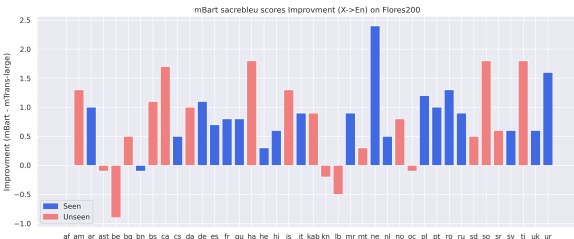

Figure 7: Performance comparison between fine-tuned mBart50 and mT-large on X2En directions.

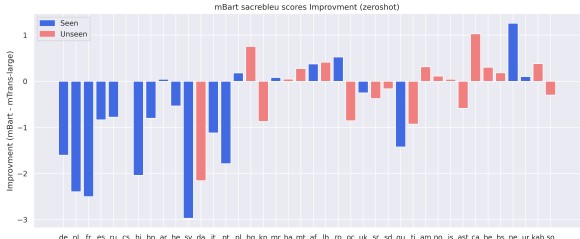

Figure 8: Performance comparison between fine-tuned mBart50 and mT-large on ZS directions. Note that we show average ZS sacrebleu scores in this Figure, e.g: de column denotes $All \leftrightarrow de$ (All ZS directions translating from and into german).

other metrics (Sacrebleu, Chrf++, Comet) in this section.

### A.5.1 Resource-level Analysis

Table 9, 10, and 11 show the Resource level analysis of the mTransformer-large model for both English-centric and Zero-shot directions across three different metrics. Combined with Table 3, we verify that our conclusions in section 5.1 are consistent across all four metrics.

|  | Sacrebleu | | | | | |
|  | Target | | | | | |
| Source | En | High | Med | Low | e-Low | Avg |
| En | - | 29.55 | 30.86 | 20.88 | 13.25 | 23.63 |
| High | 32.73 | 11.41 | 12.09 | 5.82 | 2.36 | 7.92 |
| Med | 32.65 | 11.81 | 11.95 | 6.12 | 2.62 | 8.12 |
| Low | 28.40 | 9.41 | 9.50 | 4.60 | 2.05 | 6.39 |
| e-Low | 21.00 | 7.72 | 8.15 | 4.76 | 2.18 | 5.70 |
| Avg | 28.69 | 10.09 | 10.42 | 5.32 | 2.30 | 7.02 |

Table 9: Resource-Based Translation Performance Analysis of mT-large based on Sacrebleu. We include both English-centric and zero-shot directions.

|  | Chrf++ | | | | | |
|  | Target | | | | | |
| Source | En | High | Med | Low | e-Low | Avg |
| En | - | 53.63 | 54.88 | 43.68 | 38.17 | 47.58 |
| High | 58.33 | 33.93 | 35.75 | 21.99 | 15.85 | 26.88 |
| Med | 57.34 | 33.81 | 35.07 | 22.35 | 16.22 | 26.86 |
| Low | 51.99 | 30.17 | 31.29 | 19.39 | 14.55 | 23.85 |
| e-Low | 45.46 | 28.53 | 29.92 | 21.07 | 15.66 | 23.80 |
| Avg | 53.27 | 31.61 | 33.01 | 21.2 | 15.57 | 25.33 |

Table 10: Resource-Based Translation Performance Analysis of mT-large based on Chrf++. We include both English-centric and zero-shot directions.

|  | Comet | | | | | |
|  | Target | | | | | |
| Source | En | High | Med | Low | e-Low | Avg |
| En | - | 83.58 | 82.89 | 78.77 | 70.02 | 78.82 |
| High | 84.48 | 68.37 | 69.87 | 61.29 | 50.59 | 62.53 |
| Med | 80.64 | 66.43 | 67.41 | 59.57 | 49.55 | 60.74 |
| Low | 76.56 | 62.09 | 63.57 | 54.20 | 46.08 | 56.49 |
| e-Low | 71.91 | 58.16 | 60.31 | 53.22 | 43.98 | 53.92 |
| Avg | 78.40 | 63.76 | 65.29 | 57.07 | 47.55 | 59.46 |

Table 11: Resource-Based Translation Performance Analysis of mT-large based on Comet. We include both English-centric and zero-shot directions.

## A.5.2 The impact of data and English-centric performance

Table 12, 13, and 14 show the impact of data and English-centric performance of the mTransformer-large model across three different metrics. Combined with Table 4, we verify that our conclusions in section 5.1 are consistent across all four metrics.

| Metrics | Features | Data-size$^{\dagger}$ | | En-centric perf. | |
| | | Src_size | Tgt_size | Src→En | En→Tgt |
| Correlation | Pearson | 0.15 | 0.52 | 0.40 | 0.67 |
| | Spearman | 0.18 | 0.62 | 0.38 | 0.69 |
| Regression | R-square | 30.93% | | 60.97% | |
| | MAE | 4.12 | | 3.46 | |
| | RMSE | 4.96 | | 4.16 | |

Table 12: Analysis of zero-shot performance considering data size and English-centric performance based on Sacrebleu.

| Metrics | Features | Data-size$^{\dagger}$ | | En-centric perf. | |
| | | Src_size | Tgt_size | Src→En | En→Tgt |
| Correlation | Pearson | 0.12 | 0.54 | 0.34 | 0.74 |
| | Spearman | 0.15 | 0.59 | 0.35 | 0.72 |
| Regression | R-square | 34.47% | | 64.50% | |
| | MAE | 8.12 | | 6.96 | |
| | RMSE | 10.07 | | 8.42 | |

Table 13: Analysis of zero-shot performance considering data size and English-centric performance based on Chrf++.

| Metrics | Features | Data-size$^{\dagger}$ | | En-centric perf. | |
| | | Src_size | Tgt_size | Src→En | En→Tgt |
| Correlation | Pearson | 0.25 | 0.43 | 0.53 | 0.66 |
| | Spearman | 0.31 | 0.47 | 0.46 | 0.60 |
| Regression | R-square | 4.40% | | 69.53% | |
| | MAE | 11.08 | | 7.35 | |
| | RMSE | 13.67 | | 9.04 | |

Table 14: Analysis of zero-shot performance considering data size and English-centric performance based on Comet.

## A.5.3 The effect of Linguistic properties

We investigate how zero-shot performances change if the source and target languages are linguistically more similar, considering language family and writing system. Table 15 demonstrates the fine-grained analysis regarding the effect of linguistic properties on the ZS translation quality.

| | X resource | | | | Y resource | | | |
| | eLow | Low | Med | High | eLow | Low | Med | High |
| If X and Y belong to the same Language Family | | | | | | | | |
| No | 5.05 | 5.92 | 7.67 | 7.49 | 2.12 | 4.82 | 9.77 | 9.43 |
| Yes | 9.15$^{***}$ | 8.74$^{*}$ | 9.62$^{ns}$ | 9.37$^{ns}$ | 3.14$^{*}$ | 7.69$^{***}$ | 13.16$^{**}$ | 12.88$^{**}$ |
| If X and Y use the same Writing system | | | | | | | | |
| No | 4.86 | 5.10 | 6.78 | 6.87 | 1.58 | 3.97 | 9.31 | 8.67 |
| Yes | 6.97$^{***}$ | 9.15$^{***}$ | 9.56$^{***}$ | 9.66$^{***}$ | 3.21$^{***}$ | 8.13$^{***}$ | 11.71$^{***}$ | 12.68$^{***}$ |

Welch's t-test: $^{***}p <= 0.001$, $^{**}0.001 < p <= 0.01$, $^{*}0.01 < p <= 0.05$, ns denotes $0.05 < p$

Table 15: The impact of linguistic properties on zero-shot performance. To investigate it in depth, we analyze it in fine-grained levels by observing different resource levels of Y. We also conducted Welch's t-test to validate if one group is significantly better than another.

## A.5.4 Overall Correlation Analysis using all factors

Our findings hold consistently across various evaluation metrics, spanning word, sub-word, character, and representation levels. For analyses in the section 5.2 and 5.3, we show the additional results that based on Sacrebleu, Chrf++, Comet in Table 16.

## A.5.5 The role of Off-Target Issue

We utilized SpBleu in Section 5.4 to align with the setup employed by Zhang et al. (2020). We

| ID | Features | R-square | MAE | RMSE |
|----|----------|----------|-----|------|
| | Sacrebleu (mT-large) | | | |
| 1 | En_performance | 60.97% | 3.46 | 4.16 |
| 2 | 1 + Vocab-Sim | 79.09% | 2.85 | 3.62 |
| 3 | 2 + Linguistic-features | 81.27% | 2.80 | 3.54 |
| | Chrf++ (mT-large) | | | |
| 4 | En_performance | 64.50% | 6.96 | 8.42 |
| 5 | 4 + Vocab-Sim | 77.22% | 6.87 | 8.49 |
| 6 | 5 + Linguistic-features | 84.45% | 5.74 | 7.09 |
| | Comet (mT-large) | | | |
| 7 | En_performance | 69.53% | 11.08 | 13.67 |
| 8 | 7 + Vocab-Sim | 77.22% | 6.87 | 8.49 |
| 9 | 8 + Linguistic-features | 79.51% | 6.63 | 8.30 |

Table 16: Prediction of Zero-Shot Performance using En-Centric performance, vocabulary overlap, and linguistic properties.

further provide the correlation results (correlation coefficient) based on Chrf++ and Sacrebleu below. Note that all experimental setups are the same, the only difference is the change of MT metric, for more details please check the setup descriptions in Section 5.4.

| | Sacrebleu | Chrf++ | SpBleu |
|---|-----------|--------|--------|
| Spearman | -0.07 | -0.07 | -0.08 |
| Pearson | -0.06 | -0.03 | -0.07 |

Table 17: Correlation coefficient between zero-shot performance and off-target rate when focusing on directions where the off-target rate is considerably low (less than 5%).

## A.6 Performance for all directions

Figure 9 and 10 illustrates the specific results of the mTransformer-large model on all 1,560 zero-shot directions using four metrics.

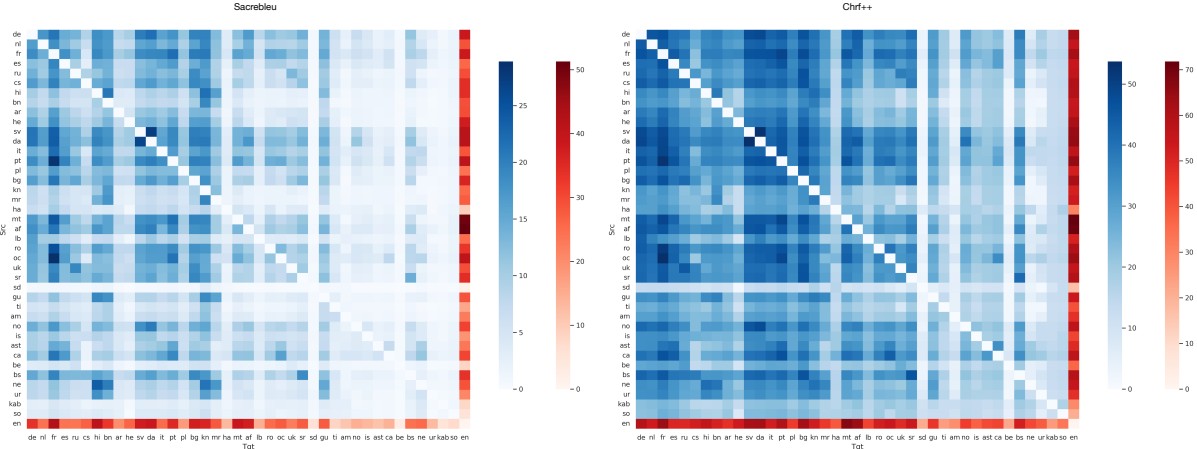

Figure 9: Zero-shot performance of mTransformer-large on 1560 directions for Sacrebleu and Chrf++

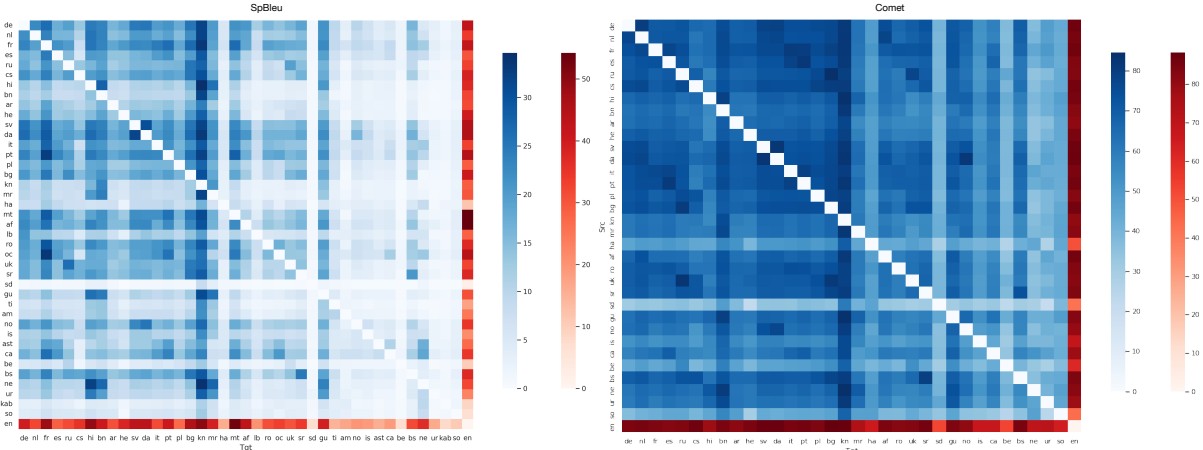

Figure 10: Zero-shot performance of mTransformer-large on 1560 directions for SpBleu and Comet