# OpenReview forum: "Towards a Better Understanding of Variations in Zero-Shot Neural Machine Translation Performance"
_EMNLP/2023/Conference — EMNLP 2023 Main_

### Official Review · Reviewer_3cCd · 2023-08-04

**Soundness:** 3

**Excitement:**

3: Ambivalent: It has merits (e.g., it reports state-of-the-art results, the idea is nice), but there are key weaknesses (e.g., it describes incremental work), and it can significantly benefit from another round of revision. However, I won't object to accepting it if my co-reviewers champion it.

**Paper Topic And Main Contributions:**

This paper focuses on the phenomenon of variations in zero-shot multilingual neural machine translation performance, and investigate the factors behind the phenominon via extensive experiments. Since the current MNMT datasets have limited training sentences and exhibit high levels of imbalance, the authors first introduce a new MNMT dataset EC40 which contains 40 non-English languages with different linguistic properties and all directions for further research. Experiments show that zero-shot directions have much lower performance and higher variations. To explain the high zero-shot variations, the authors identify three key factor: English-centric translation capacity, vocabulary overlap and linguistic properties, Among which the capacity of out-of-English translation serves as the most important one. Moreover, the authors argue that the off-target problem is a symptom of inadequate zero-shot performance rather than the root cause. Based the observations. the authors finally give some potential remedies to enhance the zero-shot performance of MNMT.

**Questions For The Authors:**

- Why do translations from/to medium-resource languages have better quality than that from/to high-resource languages according to Table 3?
- While the paper notes that high variation is not the sole cause of low average zero-shot performance, the relationship between these factors and the root cause of low performance is not discussed in depth. Could the authors provide further insights into the underlying causes of low performance and how they relate to the factors of variation?

**Reasons To Accept:**

- This paper delves into the phenomenon of high variations in zero-shot translation performance, and design extensive and comprehensive experiments to demonstrate the factors of the phenomenon. The conclusions are general regardless of the resource level of the source and target language.
- A new multilingual translation dataset is released for community which is suitable for academic research.
- The authors propose a new point of view that the off-target problem is a symptom rather than the root cause since directions with low off-target rate may also have poor performance.

**Reasons To Reject:**

- The three factors of variations are intuitive and have already been obeserved in previous works. No novel phenomena were found.
- High variation is not the cause for low average zero-shot performance. Their relationship and the root cause of low performance are not discussed in depth.

**Reproducibility:**

5: Could easily reproduce the results.

**Reviewer Confidence:**

4: Quite sure. I tried to check the important points carefully. It's unlikely, though conceivable, that I missed something that should affect my ratings.

**Typos Grammar Style And Presentation Improvements:**

- "bleu" -> "blue" in the caption of Table 3
- Use vector graphics for figures

---

> ### Author Rebuttal · Authors · 2023-08-27
>
> Dear reviewer, we would like to thank you for your time, effort, and thorough reviews! We are really encouraged by the reviews that highlight our work:
>
> 1. Design extensive and comprehensive experiments to delve into the phenomenon of high variations in zero-shot MT.
> 2. The conclusions are general regardless of the resource level of languages.
> 3. A new dataset is suitable for academic research.
> 4. Propose a new point of view that the off-target problem is a symptom.
>
> We have responded to your comments below and updated our revision accordingly. If you still have additional questions and suggestions, we would be happy to answer your questions and incorporate the suggestions into the paper.
>
> > **Reasons To Reject 1.** The three factors of variations are intuitive and have already been observed in previous works. No novel phenomena were found.
>
> We respectfully hold a different perspective on this matter. To the best of our knowledge, this paper represents the first attempt to comprehensively and systematically study in depth the impact of these three factors on zero-shot translation qualities. **Our approach distinctly sets us apart from previous research, and we invite you to assess the recent relevant works that we have collated for comparison (all these references have also been introduced in our paper).**
>
> | Paper                                                    | Task              | Descriptions                                                                                              |
> | -------------------------------------------------------- | ----------------- | --------------------------------------------------------------------------------------------------------- |
> | Zhang et al. (2020) | Zero-Shot NMT     | Investigates how modeling capacity affects ZS performance                                                                   |
> | Wu et al. (2021); Raganato et al., (2021); Wang et al. (2021) | Zero-Shot NMT     | MNMT models are prone to forget the language label information  |
> | Chen et al. (2022); Gu et al. (2019); Tang et al. (2021); Wang et al. (2021) | Zero-Shot NMT     | Examines the impact of model initialization on ZS performance|
> | Pan et al., 2021; Ji et al., (2020); Wu et al. (2021); | Zero-Shot NMT  | Recognize the inconsistency of semantic representation in different languages |
> | Aharoni et al. (2019) | Zero-Shot NMT     | Suggests that adding more languages in an MNMT system improves zero-shot performance more for close language pairs |
> | Pires et al., (2019)  | Cross-lingual NLU | Concludes that transfer is more successful for languages with high lexical overlap and typological similarity |
> | Lauscher et al., (2020) | Cross-lingual NLU | Concludes that transfer is more successful for syntactically or phonologically similar languages |
>
> **We would like to reiterate the novelty and contribution of our research concerning the three factors, and we invite you to substantiate your claims with references for a more profound discussion.**
>
> 1. Our identification of the considerable influence of out-of-English translation quality on zero-shot variation is novel, supported by detailed insights provided in lines 566 to 574. We further provide insight into how to use this finding to improve the zero-shot NMT in Section 5.4.
> 2. We found that performance in the close language pairs (with same linguistic properties) is significantly better than distant pairs regardless of resource level, and this phenomenon is especially pronounced for smaller models. We further stress the importance of focusing efforts on distant language pairs rather than exclusively concentrating on resource-rich pairs, as prior ZS-NMT research has done (lines 575--586).
> 3. We have consistently observed that vocabulary overlap plays a significant role in explaining zero-shot variations, encouraging greater cross-lingual transfer and knowledge sharing via better vocabulary sharing has the potential to enhance zero-shot translations (lines 587--600).
>
>
> ---
>
>
> > **Question 1.** Why do translations from/to medium-resource languages have better quality than that from/to high-resource languages according to Table 3?
>
> Thank you for raising this point. This phenomenon is observed in other En-centric MNMT experiments, as evidenced by the studies conducted by [1] and [2].
>
> Furthermore, our own experiments have shown that certain medium-resource languages, like Afrikaans, demonstrate the highest performance across all metrics. This exceptional performance of Afrikaans contributes significantly to raising the overall average score within the medium-resource group. This perspective is consistent with earlier research, which highlights Afrikaans' extensive lexical and syntactic borrowings from English, as well as its close linguistic affiliations with multiple neighboring languages of English [3].
>
> ---
> > **Reasons To Reject 2 and Question 2** High variation is not the cause for low average zero-shot performance. Their relationship and the root cause of low performance are not discussed in depth. While the paper notes that high variation is not the sole cause of low average zero-shot performance, the relationship between these factors and the root cause of low performance is not discussed in depth. Could the authors provide further insights into the underlying causes of low performance and how they relate to the factors of variation?
>
> Thank you for your thoughtful feedback. We acknowledge the universally recognized challenge of zero-shot performance consistently falling behind supervised performance in the broad domain of deep learning. The community as a whole is deeply engaged in understanding this issue and actively works to enhance it. In the context of Multilingual Machine Translation, **recent research has focused on seeking explanations for the causes of overall low ZS performance and trying to enhance it.** Please refer to the literature and descriptions we demonstrated in *Reasons To Reject 1* and Sec.2.2 in the paper.
>
> **In contrast, our study introduces a fresh perspective within zero-shot NMT: the presence of high variations.** This implies that ZS directions don't uniformly suffer from low performance; instead, some directions exhibit decent results. Specifically, it is evidenced by certain ZS directions exhibit promising performance, with approximately 70% of ZS-condition-1 achieving SpBleu scores surpassing 20 (see Figure 1 on page 1). In stark contrast, merely 10% of the remaining directions achieve comparable SpBleu scores. This intriguing phenomenon prompted us to further understand the underlying factors driving this extensive variation in zero-shot NMT.
>
> We recognize that our investigation of high variations in zero-shot performance adds an important layer of insight to the discourse surrounding zero-shot NMT, which provides an additional perspective than understanding the root causes of overall poor performance in zero-shot scenarios. Drawing from our findings, we have identified three key factors responsible for driving these variations and have subsequently offered insights to enhance zero-shot NMT performance. Of note, our conclusions emphasize the significance of resource-lean and distant language pairs as a focus for improving zero-shot performance. Crucially, this insight aligns with, rather than contradicts, existing efforts aimed at improving performance in resource-rich language pairs, while also addressing the aspect of language relatedness that has often been overlooked.
>
>
> ---
>
> References:
>
> [1] Zhang, Biao, et al. "Improving massively multilingual neural machine translation and zero-shot translation."
>
> [2] Yang, Yilin, et al. "Improving multilingual translation by representation and gradient regularization."
>
> [3] Zhou, Zhong, and Alex Waibel. "Family of origin and family of choice: Massively parallel lexiconized iterative pretraining for severely low resource text-based translation."

---

### Official Review · Reviewer_2RvR · 2023-08-04

**Soundness:** 3

**Excitement:**

3: Ambivalent: It has merits (e.g., it reports state-of-the-art results, the idea is nice), but there are key weaknesses (e.g., it describes incremental work), and it can significantly benefit from another round of revision. However, I won't object to accepting it if my co-reviewers champion it.

**Paper Topic And Main Contributions:**

This paper conducts a multilingual dataset (EC40), which contains 66M sentence pairs in total over 40 languages, by expanding the OPUS dataset.
Further, the authors adopt EC40 as the train set to explore the influence of three factors (English-centric translation capacity, vocabulary overlap, and linguistic properties) on the zero-shot translation.
Besides, the authors

**Questions For The Authors:**

Please see 'Reasons to reject'

**Reasons To Accept:**

1. The Authors conduct a large multilingual translation dataset, containing 66M sentence pairs in total.
2. This paper tries to explore the influence of three aspects on the performance of zero-shot translation.
3. This paper conducts expensive experiments to prove their conclusion.

**Reasons To Reject:**

Although the authors conduct expensive experiments, some questions about it are as follows:

1. What is the difference between Sacrebleu and SpBleu in the experiments? And are their scores in different translation directions comparable?
2. Why EC40 is balanced, while EC40's scale is from 50K to 5M and the Script of most languages is Latin?
3. Why the m2m-100 model could be evaluated directly, without any fine-tuning? No gap between itself train set and EC40?
4. What is referred to in the CV metric for zero-shot in Table 1? And why the highest score is noted by underline?
5. What is the inner relationship among these aspects (English-centric capacity, vocabulary overlap, linguistic properties, and off-target issue)
6. Why switch the metric to 'Sacrebleu' in Sec.5.4?

**Reproducibility:**

4: Could mostly reproduce the results, but there may be some variation because of sample variance or minor variations in their interpretation of the protocol or method.

**Reviewer Confidence:**

4: Quite sure. I tried to check the important points carefully. It's unlikely, though conceivable, that I missed something that should affect my ratings.

---

> ### Author Rebuttal · Authors · 2023-08-27
>
> Dear reviewer, we would like to thank you for your time, effort, and thorough reviews! We are really encouraged by the reviews that highlight our work:
>
> 1. Conduct a large multilingual translation dataset.
> 2. Explore the influence of factors on the performance of zero-shot translation.
> 3. Conduct extensive experiments.
>
> We have responded to your comments below and updated our revision accordingly. If you still have additional questions and suggestions, we would be happy to answer your questions and incorporate the suggestions into the paper.
>
> ---
>
> > **Reasons To Reject 1.** What is the difference between Sacrebleu and SpBleu in the experiments? And are their scores in different translation directions comparable?
>
> Thank you for the question. To address your question, it's important to note that our paper does not involve cross-metric comparisons. Secondly, our approach focuses on providing a comprehensive analysis, incorporating various metrics to ensure a thorough evaluation. Specifically, we incorporate the following metrics:
> * ChrF++ (character level)
> * SpBleu (tokenized sub-word level)
> * SacreBleu (detokenized word level)
> * COMET (representation level)
>
> Our intention in employing these multiple metrics is to ensure a holistic assessment of our experiments. Notably, we maintain consistency in our findings across all these metrics, which lends robustness to our conclusions. To delve further into the analysis based on COMET, we kindly direct you to our responses in the context of *"Reasons To Reject 5"* where more detailed insights are provided, specifically addressing the concerns raised by reviewer jc8P.
>
> ---
>
> > **Reasons To Reject 2.** Why EC40 is balanced, while EC40's scale is from 50K to 5M and the Script of most languages is Latin?
>
> Thank you for raising this point. We pointed out that EC40 is balanced across resources and languages (lines 218--221) instead of the number of training sentences among the whole dataset or type of scripts. Please check Sec.A.1 for more details in the Appendix. Specifically, the EC40 is carefully balanced across various scales (from lines 234-240):
>
> 1. For each resource, the number of languages is the same (10 languages).
> 2. For each resource, the number of training sentences is balanced (line 238-240), e.g.: 5M for all High-resource languages)
> 3. For each language family, we incorporate 8 languages (lines 236-238).
>
> ---
>
> > **Reasons To Reject 3.** Why the m2m-100 model could be evaluated directly, without any fine-tuning? No gap between itself train set and EC40?
>
> Thank you for your question. **Our goal in introducing the m2m-100 models is to investigate whether the high variations in the ZS performance phenomenon hold across models.** Specifically, m2m-100 is a trained multilingual translation model using 7.5B parallel sentences covering 2200 training directions, whereas our EC40 contains only 66M sentences. Moreover, m2m-100 is a strong translation model achieving SOTA results in multiple translation directions including both supervised and zero-shot ones [1].
>
> **An additional noteworthy point is the alignment of domain factors.** Our validation and test sets encompass data from diverse domains, a characteristic shared with the m2m-100 model's training data, which includes content from multiple domains such as CCMatrix. This feature contributes to a smaller domain mismatch between the m2m-100 model and our test set, further justifying our direct evaluation approach.
>
> ---
>
> > **Reasons To Reject 4.** What is referred to in the CV metric for zero-shot in Table 1? And why the highest score is noted by underline?
>
>
> Thank you for your question. **In the context of Table 1, the reference is to the Coefficient of Variation ($CV = \frac{\sigma}{\mu}$), which serves as a metric used to quantify the magnitude of variation in zero-shot translation performances when compared to supervised translation directions.** The purpose of utilizing this metric is outlined in lines 349--355 of the paper. Specifically focusing on Table 1, the Coefficient of Variation (CV) scores are computed for various machine translation (MT) metrics, namely Sacrebleu, Chrf++, and Spbleu. These scores are calculated under different settings, such as language pairs denoted as En→X.
>
>
> Notably, in Table 1, the highest CV score for a particular metric and setting is indicated by being underlined. This visual emphasis (underline) draws attention to the translation scenarios where the variation between zero-shot directions is more pronounced than supervised ones, helping to highlight significant findings of high variations.
>
>
>
> ---
>
> > **Reasons To Reject 5.** What is the inner relationship among these aspects (English-centric capacity, vocabulary overlap, linguistic properties, and off-target issue)
>
> Thank you for this question. **While investigating correlations between these factors could indeed provide valuable insights, the primary objective of our study is to uncover their relationships with our target variable: zero-shot translation ability.** The main focus of the paper is to shed light on how these factors interpret high variations in zero-shot translation performance.
>
> Moreover, we discuss one intuitively strong relationship here: One of the intuitive connections lies between vocabulary overlap and linguistic properties. For instance, languages that share the same writing system might exhibit a higher vocabulary overlap due to lexical similarities. This part is explicitly explored in the paper (lines 382--488), highlighting that vocabulary overlap serves as a more fundamental indicator of surface-level similarity compared to more intricate linguistic metrics such as language family or typology distance.
>
> ---
>
> > **Reasons To Reject 6.** Why switch the metric to 'Sacrebleu' in Sec.5.4?
>
> Thank you for your question. I assume you mean why use Sacrebleu in Sec.5.3 instead of Sec.5.4? Please let us know if this is not true.
>
> We utilized Sacrebleu in Sec. 5.3 to align with the setup employed by [2] (the paper hypothesized the off-target issue is the root cause of the low ZS performance). we further provide the correlation results (correlation coefficient) based on Chrf++ and Spbleu below. Note that all experimental setups are the same, the only difference is the change of MT metric, for more details please check the setup descriptions in Sec.5.4.
>
> |           | Sacrebleu | Chrf++ | SpBleu|
> |-----------|-----------|--------| ------|
> | Spearman  |  -0.07    | -0.07  | -0.08 |
> | Pearson   |  -0.06    | -0.03  | -0.07 |
>
> ---
>
>
> References:
>
> [1] Fan, Angela, et al. "Beyond english-centric multilingual machine translation."
>
> [2] Zhang, Biao, et al. "Improving massively multilingual neural machine translation and zero-shot translation."

---

### Official Review · Reviewer_jc8P · 2023-08-04

**Soundness:** 3

**Excitement:**

3: Ambivalent: It has merits (e.g., it reports state-of-the-art results, the idea is nice), but there are key weaknesses (e.g., it describes incremental work), and it can significantly benefit from another round of revision. However, I won't object to accepting it if my co-reviewers champion it.

**Missing References:**

All the references are appropriate.


**Paper Topic And Main Contributions:**

This paper presents an in-depth study and analysis of zero-shot neural machine translation (ZS-NMT). Starting from the assumption that knowledge sharing is the basement for ZS-NMT, this paper runs an extensive analysis with the aim of better understanding zero-shot translation and its limit and benefit. This study reveals that there are high variations in performance across different language directions. The reasons for this can be ascribed to the role of English-centric translation capability, vocabulary overlap between source and target languages, and linguistic properties. The analysis shows that the first factor is the most important one. To have a fair analysis, a novel dataset for training purposes is proposed and released. This dataset balances data distribution and considers linguistic characteristics of diverse representations of languages.



**Questions For The Authors:**

-) Page 3 lines 238-240: What does this sentence mean?

-) Page 4 lines 266-269: Is the cleaning step performed before or after selecting the data in the EC40 dataset? If it is performed after the creation of the dataset, how can you guarantee that the data distribution is not modified?

-) What are the numerator and denominator in the CV equation?

-) Page 7, why do the authors use RMSE in Table 6 but only R-square and MAE in Table 4? What is the advantage of adding RMSE in Table 6?



**Reasons To Accept:**

The paper addresses an interesting and important issue for the machine translation community. In particular, it tries to better understand zero-shot translation by running an extensive set of experiments covering a large number of languages and conditions.

The experiments take into account several factors trying to limit biases and situations that can alter the results and the findings. The overall experimental setting is grounded. The results confirm the claims being made.

I appreciate Section 5.4 where the main findings of the paper are summarized and possible future directions are identified.

The novel dataset for training purposes is an important contribution of this paper.

**Reasons To Reject:**

The weaknesses of the paper are:

1) Although the relevant papers are cited, some of the key concepts used in the paper are not properly introduced. IMHO, this reduces the target audience of the paper only to experts in multilingual neural machine translation and makes the paper difficult to follow and understand. For instance:
1.1) “issue of spurious correlation”: this is mentioned in Section 2.2 and re-used in several parts of the paper.
1.2) “off target issue”: this is mentioned for the first time at the end of the introduction.
1.3) “correlation and regression analysis”: this is mentioned in Section 5.1 and used in the remaining sections.
1.4) “sentence level off target rate”: used in Section 5.3.
All these concepts should be properly introduced in the paper.

2) The appendix can be used to report “anonymised related work (see above), preprocessing decisions, model parameters, feature templates, lengthy proofs or derivations, pseudocode, sample system inputs/outputs, and other details that are necessary for the exact replication of the work described in the paper” (see the call for papers) and the paper should be self-contained. In this paper, some relevant results are reported in the Appendix:
2.1) Section 4 “Does pre-training matter?”
2.2) Section 4 “Quantifying variation”
2.3) Section 5.2.1” “Linguistic Properties”
This makes the paper difficult to follow.

3) The vocabulary overlap part is an important contribution to the paper, but its setting is not clear. In the abstract, it is mentioned the vocabulary overlap between source and target languages, but in Section 5.2.2 lines 509-513 it is not clear what correlation is computed and it seems to refer only to the source language. This part should be clarified and improved.

4) Some of the results are somehow expected, in particular, performance improvements when source and target languages adopt the same writing script. Can this part be moved in the Appendix using the left space for improving other parts?

5) The MT evaluation is based on three metrics: Bleu, Chrf++, and SpBleu. These are all string-matching metrics. I was wondering if the use of COMET could make the findings more robust. Not sure, if COMET covers all the languages involved in this study. If not, it can be used only for a subset of language directions.

**Reproducibility:**

4: Could mostly reproduce the results, but there may be some variation because of sample variance or minor variations in their interpretation of the protocol or method.

**Reviewer Confidence:**

4: Quite sure. I tried to check the important points carefully. It's unlikely, though conceivable, that I missed something that should affect my ratings.

**Typos Grammar Style And Presentation Improvements:**

Page 7 line 533: “model capacities. And it shows” dot should be replaced with a comma.

The paper is quite dense with several important concepts that are not fully introduced (see issue 1 in the weaknesses section) or relevant results for the paper that are in the Appendix (see issue 2 in the weaknesses section). To favor the understanding and reading of the paper, a drastic decision has to be made by moving some experiments to the Appendix obtaining more space for improving the remaining parts. A good candidate can be Section 5.3.

---

> ### Author Rebuttal · Authors · 2023-08-27
>
> Dear reviewer, we would like to thank you for your time, effort, and thorough reviews! We are really encouraged by the reviews that highlight our work:
>
> (1) Addresses an interesting and important issue for the machine translation community by providing in-depth study and extensive analyses;
>
> (2) The experimental setting is grounded and the results confirm the claims being made;
>
> (3) The main findings of the paper are summarized and possible future directions are identified;
>
> (4) The novel dataset is an important contribution.
>
> We have responded to your comments below and updated our revision accordingly. If you still have additional questions and suggestions, we would be happy to answer your questions and incorporate the suggestions into the paper.
>
> ---
>
> > **Reasons To Reject 1, 2, and 4.** Some of the key concepts used in the paper are not properly introduced. Some relevant results are reported in the Appendix. Can this part be moved in the Appendix?
>
> Thank you for offering this guidance. We have already improved the presentation of them accordingly in our revision. We really appreciate your feedback and we believe this strengthened our paper.
>
> ---
>
> > **Reasons To Reject 3.** The vocabulary overlap part is an important contribution to the paper, but its setting is not clear. In Section 5.2.2 lines 509-513, it is not clear what correlation is computed and it seems to refer only to the source language. This part should be clarified and improved.
>
> We thank you for the feedback. As outlined in line 495, in the whole paper, we measure vocabulary overlap in the same way, i.e., the extent of shared subwords between the source and target languages.
>
> When solely studying whether a higher overlap correlates with improved ZS performance (lines 503-509), we fixed the target language. e.g., the target language is German, the source side is other 39 languages; then we compute the correlation score using these 39 overlap scores and corresponding 39 ZS performances. We report the average correlation scores (for all 40 target languages) in lines 518--522.
>
> When incorporating the vocabulary overlap feature with English translation capability (En_performance) in the regression experiment (Table 6), we use the overlap directly in the regression analysis.
>
> ---
>
> > **Reasons To Reject 5.** The MT evaluation are all string-matching metrics. If the use of COMET could make the findings more robust.
>
> Thank you for this suggestion. We have conducted the COMET evaluations and ran all analyses based on COMET's supported languages (35/41 including English). **In sum, the findings based on COMET for all sections are consistent with our previous analyses. and we believe this makes our paper more solid.** Below we provide the main evidence, and we have already added them in the revision.
>
> #### Table 10: Resource-level analysis based on Comet score. This result is consistent with Table 3 in page 5. Our main conclusion in this part remains the same: the resource level of the target language has a stronger effect on ZS translation qualities.
> |       |      | Target     |       |       |       |       |       |
> |------ |------|:----------:|-------|-------|-------|-------|-------|
> |       |      |    En      | High  |  Med  |  Low  | e-Low |  Avg  |
> | Source|  En  |     -      | 0.84  | 0.83  | 0.79  | 0.70  | 0.79  |
> |       | High |    0.84    | 0.68  | 0.70  | 0.61  | 0.51  | 0.63  |
> |       |  Med |    0.81    | 0.66  | 0.67  | 0.60  | 0.50  | 0.61  |
> |       |  Low |    0.77    | 0.62  | 0.64  | 0.54  | 0.46  | 0.56  |
> |       |e-Low |    0.72    | 0.58  | 0.60  | 0.53  | 0.44  | 0.54  |
> |       |  Avg |    0.78    | 0.64  | 0.65  | 0.57  | 0.48  | 0.59  |
>
> #### Table 11: Analysis of zero-shot performance considering data size and English-centric performance based on Comet score. The result is consistent with Table 4 on page 6. Our main conclusion in this part remains the same: Out-of-English translation quality correlates with zero-shot performance the most. We also observed the data-size feature can even harder to explain the zero-shot variations (with low R-square and high MAE, and RMSE scores).
>
> |                 |                | Src_size  | Tgt_size | Src->en          | en->Tgt  |
> |-----------------|----------------|-----------|----------|------------------|----------|
> | Correlation     | Pearson        | 0.25      | 0.43     | 0.53             | 0.66     |
> |                 | Spearman       | 0.31      | 0.47     | 0.46             | 0.60     |
>
>
> |                    |                    |  Data-size | En-centric perf. |
> |-----------------|----------------|----------------|----------------------|
> | Regression      | R-square       | 4.40%     | 69.53%   |
> |                 | MAE            | 11.08     | 7.35     |
> |                 | RMSE           | 13.67     | 9.04     |
>
> #### Table 12: Prediction of ZS Performance using En-Centric performance, vocabulary overlap, and linguistic properties. We present the result based on the mT-large and the Comet score in this table. This result is consistent with Table 6 on page 7. Our main conclusion in this section remains the same: Vocabulary overlap and linguistic features can further explain the ZS variations.
> |   ID  |          Features          | R-square |   MAE   |   RMSE  |
> |-------|---------------------------|----------|---------|---------|
> |   4   |      En_performance       | 69.53%   | 11.08   | 13.67   |
> |   5   |      4 + Vocab-Sim        | 77.22%   | 6.87    | 8.49    |
> |   6   | 5 + Linguistic-features   | 79.51%   | 6.63    | 8.30    |
>
> ---
>
> > **Question 1.** Page 3 lines 238-240: What does this sentence mean?
>
> This sentence means for each resource, we control the number of languages and the number of training sentences the same.
>
> * Are the number of languages in each resource level the same?
>     * Yes, 10 languages for each resource.
> * Are the number of languages in each language family the same?
>     * Yes, 8 languages for each family.
> * Are the number of training sentences in each resource level the same?
>     * Yes, e.g.: 5M for high-resource, and 1M for Medium-resource, and etc.
>
> ---
>
> > **Question 2.** Page 4 lines 266-269: Is the cleaning step performed before or after selecting the data in the EC40 dataset? If it is performed after the creation of the dataset, how can you guarantee that the data distribution is not modified?
>
> The cleaning step is before selecting the data in the EC40 dataset. This means all numbers of training sentences reported in Table 7 (page 12) are after the cleaning, therefore our models trained using the exact numbers we reported in Table 7. Please also check the steps we list below.
>
> Download the data from OPUS -> Do the data cleaning step -> Sample data from the cleaned set -> Learn vocabulary and train models
>
> ---
>
> > **Question 3.** What are the numerator and denominator in the CV equation?
>
> The Coefficient of Variation $CV = \frac{\sigma}{\mu}$ is defined as the ratio of the standard deviation (numerator: $\sigma$) to the mean (denominator: $\mu$) of zero-shot performance.
>
> ---
>
> > **Question 4.** Page 7, why do the authors use RMSE in Table 6 but only R-square and MAE in Table 4? What is the advantage of adding RMSE in Table 6?
>
> We add RMSE to our experiments instead of only using MAE mainly because of two points:
> * Sensitivity to Large Errors: RMSE gives more weight to larger errors due to the squaring operation.
> * Outlier Impact: Because RMSE squares the differences, it is more sensitive to outliers than MAE.
>
> Therefore, we incorporated two metrics in Table 6 for a more comprehensive analysis. We appreciate your advice and we further present the RMSE for Table 4 (see below) and updated this in the revision to make the presentation more consistent.
>
> |              |                     | Data-size | En-centric perf. |
> |--------------|---------------------|-----------|----------|
> |Regression    | R-square            | 32.54%    | 61.34%   |
> |              | MAE                 | 5.47      | 4.70     |
> |              | RMSE                | 6.62      | 5.59     |

---

### Meta-Review · Area_Chair_1mQq · 2023-09-19

**Recommendation:** 3

**Metareview:**

This paper investigates the phenomenon of zero-shot neural machine translation (ZS-NMT) performance variations across different language directions and seeks to understand the underlying factors. It introduces a new dataset, EC40, and conducts extensive experiments involving 40 languages, focusing on the influence of English-centric translation capacity, vocabulary overlap, and linguistic properties.

Pros:

* Novel Dataset and In-Depth Analysis: The introduction of the EC40 dataset is highlighted as an important contribution to the paper. The extensive experiments conducted on this dataset offer valuable insights into ZS-NMT performance variations.

* Thorough Investigation: The paper explores the factors contributing to ZS-NMT performance variations comprehensively, considering English-centric translation capacity, vocabulary overlap, and linguistic properties. This in-depth analysis provides a holistic understanding of the issue.

* Identification of Off-Target Issue: The paper suggests that the off-target issue is a symptom rather than the root cause of inadequate ZS-NMT performance, offering a novel perspective on the problem.


Cons:

* Concept Clarification: Reviewer 1 points out that some key concepts used in the paper, such as "spurious correlation," "off-target issue," "correlation and regression analysis," and "sentence-level off-target rate," are not properly introduced, making the paper challenging to follow for non-experts.

* Metric Choice: Reviewer 2 questions the use of string-matching metrics (Bleu, Chrf++, and SpBleu) and suggests considering the use of COMET, although it's noted that COMET may not cover all the languages involved in the study.

* Lack of Novel Phenomena: Reviewer 3 notes that while the paper delves into ZS-NMT variations, the three identified factors are intuitive and have been observed in previous works. There is a need for more exploration of the relationship between high variation and low average ZS performance.

In summary, the paper presents a comprehensive analysis of ZS-NMT performance variations, supported by a new dataset, EC40. While it makes valuable contributions to the field, addressing the need for concept clarification, handling appendix content, and further exploring the relationship between high variation and low performance could enhance the paper's quality and impact.

---

### Decision · Program_Chairs · 2023-10-07

**Decision:**

Accept-Main

**Comment:**

This paper investigates the phenomenon of zero-shot neural machine translation (ZS-NMT) performance variations across different language directions and seeks to understand the underlying factors. It introduces a new dataset, EC40, and conducts extensive experiments involving 40 languages, focusing on the influence of English-centric translation capacity, vocabulary overlap, and linguistic properties.

Pros:

* Novel Dataset and In-Depth Analysis: The introduction of the EC40 dataset is highlighted as an important contribution to the paper. The extensive experiments conducted on this dataset offer valuable insights into ZS-NMT performance variations.

* Thorough Investigation: The paper explores the factors contributing to ZS-NMT performance variations comprehensively, considering English-centric translation capacity, vocabulary overlap, and linguistic properties. This in-depth analysis provides a holistic understanding of the issue.

* Identification of Off-Target Issue: The paper suggests that the off-target issue is a symptom rather than the root cause of inadequate ZS-NMT performance, offering a novel perspective on the problem.


Cons:

* Concept Clarification: Reviewer 1 points out that some key concepts used in the paper, such as "spurious correlation," "off-target issue," "correlation and regression analysis," and "sentence-level off-target rate," are not properly introduced, making the paper challenging to follow for non-experts.

* Metric Choice: Reviewer 2 questions the use of string-matching metrics (Bleu, Chrf++, and SpBleu) and suggests considering the use of COMET, although it's noted that COMET may not cover all the languages involved in the study.

* Lack of Novel Phenomena: Reviewer 3 notes that while the paper delves into ZS-NMT variations, the three identified factors are intuitive and have been observed in previous works. There is a need for more exploration of the relationship between high variation and low average ZS performance.

In summary, the paper presents a comprehensive analysis of ZS-NMT performance variations, supported by a new dataset, EC40. While it makes valuable contributions to the field, addressing the need for concept clarification, handling appendix content, and further exploring the relationship between high variation and low performance could enhance the paper's quality and impact.